# Life and Family Values Similarity in Inter-Ethnic and Inter-Faith Couples

**DOI:** 10.3390/bs10010038

**Published:** 2020-01-19

**Authors:** Elena Yu. Chebotareva, Marina I. Volk

**Affiliations:** 1Moscow State University of Psychology and Education, National Research University “Higher School of Economics”, Ulitsa Sretenka, 29, 127051; Ulitsa Myasnitskaya, 20, 101000 Moscow, Russia; 2Peoples’ Friendship University (RUDN University), Ulitsa Miklukho-Maklaya, 6, 117198 Moscow, Russia; volkmarin@gmail.com

**Keywords:** life values, family values, inter-ethnic marriage, inter-confessional marriage, marital satisfaction, marital adjustment

## Abstract

Inter-cultural families are an integral part of modern society, the institution of mutual influence of different cultures, of a person’s identity transformation. The studies of marital adjustment, values, and attitudes consistency in inter-cultural couples provide contradictory results. To resolve contradictions in this area, comparative studies of inter-cultural families of different types are important. The aim of the study is the comparative analysis of life and family values in inter-cultural couples, differing by the ethnic and religious affiliation of spouses. The participants: 69 couples: 20 mono-ethnic Russian, 30—inter-ethnic, inter-faith (Russians/Arabs); 19—inter-ethnic, with a common religion (Russian/Transcaucasian, Christians). The methods: Value Survey (Schwartz), Marital Role Expectations and Aspirations (Volkova), Marital Satisfaction Test (Stolin et al.), Mann–Whitney U-test, paired T-test. In inter-cultural couples, the spouses’ life-values coincidence is significantly less strong than in mono-cultural ones. However, in couples with common religious differences, their life values reflect not so much the contradictions, but the complementarity of traditional gender values. In general, the different cultures of spouses (both ethnic and religious) optimizes the process of comparing values and family attitudes. Despite a number of difficulties, spouses from inter-cultural couples generally have more consistent ideas about their family life.

## 1. Introduction

Inter-cultural families have become an integral part of society, their number is steadily growing all over the world [1]. The wave of migration in recent years has drawn increased attention to the study of the processes of integration of migrants into the host community; including the psychological problems of inter-cultural families and their influence on the identities of spouses. Many researchers of the problem point out that religious identity is the main barrier for inter-ethnic marriage, although inter-faith marriage is becoming more common [2,3,4,5]. In general, studies of marital adjustment and satisfaction with relationships in inter-cultural couples provide contradictory results. Some studies show that, in cross-cultural couples, different cultural traditions and values of spouses cause great conflicts and, accordingly, these couples are less satisfied with their marriages [6,7]. Many studies show that homogeneous marriages are more prosperous and stable [8,9]. Other studies have confirmed an equal level of satisfaction and sustainability of inter-cultural and mono-cultural marriages [10]. But there is also some scientific evidence that satisfaction with marriage in inter-cultural families may be greater than in mono-cultural ones [11]. Among the factors that help couples to cope with the difficulties of inter-cultural marriages identified in the studies are the positive ethnic identity of the spouses, a positive attitude towards the culture of the partner [7,12].

Many researchers of inter-cultural marriages agree that one of the most acute problems is the comparison of values and attitudes of the spouses. There is a wide range of points of view on this issue. On the one hand, it is assumed that inter-cultural families are mainly created by people who either have already culturally assimilated or have certain values, such as universalism, tolerance, and multiculturalism [13]. On the other hand, it is assumed that the similarity of the spouses’ worldviews is the main problem of such marriages [13,14]. Somewhere in the middle of this continuum, there are ideas that cultural differences can be reduced due to the similarity in education and personal development [15] or that two people create their own “relational culture” that does not contradict the cultural values of each spouse [10,16]. Thus, it is important to study the life values of spouses in inter-cultural marriages, taking into account their family values and attitudes towards family life [17]. In many studies of inter-cultural marriages, the variable of religious affiliation of spouses is not controlled, which may explain the inconsistency of data from different studies in this area.

We have conducted an empirical study with the aim of a comparative analysis of life and family values in inter-cultural families, which differ only in in ethnic or in ethnic and religious affiliation of spouses. We also pursued the aim of assessing differences in the degree of general marital satisfaction of the spouses in different types of inter-cultural couples. That allows us to understand better the psychological mechanisms for the implementation of cultural and personal values in family relationships; about the processes of mutual adaptation (and acculturation) in inter-cultural families.

In this study of life values, we focus on the S. Schwartz approach, who defines basic values as the desired cross-situational goals, varying in importance and serving as guiding principles in human life [18]. S. Schwartz regards culture as a latent, hypothetical variable that can be measured only through its manifestations, in particular, through life values. He also argues that ethnic, professional, religious, and other groups within society can be exposed to different cultural influences and can develop different value preferences. These differences lead to social tension, conflict, and change. According to S. Schwartz’s theory, the person’s values exist at two levels: at the level of normative ideals and at the level of individual priorities. The first level is more stable and reflects the person’s ideas about how to act, thereby determining his life principles of behavior. The second level is more dependent on the external environment, for example, on group pressure and correlates with specific actions of a person [19,20]. Family values, accordingly, we consider as cultural values related to the sphere of family life, in particular, to the structure, functions, roles, beliefs and ideals of the family.

In accordance with these ideas, we find it important to take into account both ethnic and religious factors when considering the values of spouses from inter-cultural families. As P. Tillich stated, religion and culture have an intimate reciprocal relationship and a long history of interaction, he said that culture may be considered as the form of religion and religion as the substance of culture [21]. Religion and ethnos are a specific type of social associations of people, not always closely interlinked with each other. In the ethnic processes taking place in the world, the role of religion is great. However, ethnic groups differ in the degree of their conjugation with a faith. Many people are grounded in mainstream religious traditions, belonging to one of the world’s main faith. However, some other people belong to small, non-mainstream groups that “may be insular and extreme–and sometimes intolerant, legalistic, and fanatic. … These groups may be considered fundamentalist or radical” [22] p. 1092. Accordingly, to identify the general patterns of inter-ethnic and inter-faith marriages, we should focus on world’s main religions and on the ethnic groups conjugated with a faith. In the study of the role of religion in family relationships, it is also important to take into account the parameter of the spouses’ religiosity degree. Most often, a self-reporting scale is used to assess the degree of religiosity with atheism at one end and extreme religiosity at the other end of the scale [23]. The assessment is based on such parameters as faith in the immortality of the soul, in the existence of a higher power, a supernatural world, as well as the importance of these issues for the respondent [24].

## 2. Materials and Methods

The study involved 138 people (69 couples). Twenty couples were with both spouses ethnic Russians, belonging to Orthodox Christianity; 30 couples consisted of Russians, Orthodox Christian women and men from various Arab countries (Yemen, Lebanon, Syria, Jordan, Palestine, etc.), belonging to Islam; 19 couples consisted of Russian, Orthodox Christian women and their husbands from the Transcaucasian region (mostly Armenians and Ossetians and some Georgians, all men were Christians;). Armenian people belong to Armenian Apostolic Church, which have a number of features in dogma and rituals that distinguish it from Byzantine Orthodoxy and, accordingly, from the Russian Orthodox Church. But the well-known Italian orientalist—Armenologist Giovanni Guita—believes that, despite a number of theological differences, if “faith is stated purely theoretically, the differences between Catholicism, Byzantine-Slavic Orthodoxy and the Armenian Church are minimal, the commonality is, conditionally speaking, 98 or 99 percent” [25]. What is important for this research aims, is that both churches preach the same life values. North Ossetia is often mentioned as an Orthodox republic, due to the very strong orientation of the population to the Orthodox religion and culture. The Constitution of South Ossetia enshrines the provision according to which “Orthodoxy is the basis of the worldview of the Ossetian people.” [26]. About 90% of the Georgian population profess Orthodox Christianity—most of them are adherents of the Georgian Orthodox Church, which have prayer and eucharistic communion with the Russian Orthodox Church [27].

So, two types of inter-cultural couples were analyzed: inter-ethnic–mono-faith (which we will call shortly “inter-ethnic”) and inter-ethnic, complicated by different religiosity (next we will call it shortly “inter-faith”). The control grope for both these research groups were the group of mono-ethnic couples with common religion.

These subsamples were formed from a much larger number of respondents 225 couples (450 persons), including 85 Russian—Arab; 80 Russian—Russian and 60 Russian—Caucasian. This helped us to equalize subsamples that were consistent with each other by most basic socio-demographic characteristics: age, education level, type of work, children, previous divorces, children from previous marriages, self-rating level of material well-being and so on.

The age of the respondents was from 23 to 47 years (M = 36: women—32; men—40), they have been married from 3 to 24 years (M = 6.3); for 30% of respondents this marriage was the second; the majority (87%) of the surveyed families had 1–2 children, aged from 1 to 13 years old (M = 5.2); 10% of couples had children from previous marriages, everyone has a job (full-time), mainly hired employees, ordinary employees or mid-level managers, foreign partners lived in their spouse’s country from 5 to 30 years (M = 9.2), everyone were moderately religious.

In the groups of intercultural marriages, unlike monocultural, there were no couples with unregistered marriages (in the monocultural group, 30% cohabitation); in the group of Russian–Caucasian marriages, unlike the other 2 groups, women on average occupy lower-level positions at work. We believe that the absence of cohabitation is related to cultural traditions and migration issues. A lower level of the job position of women married to a Caucasian partner is due to the factor of their living in a more traditional culture. Furthermore, in the Caucasian–Russian group the age of respondents was on average lower (M = 31) than in the other two groups. Russian–Arab (RA) and mono-cultural Russian (RR) families lived in different cities of Russia, Russian–Caucasian (RC) families lived in their native regions of their husbands—in cities of the Transcaucasian regions.

Methods of the research. To study the life values of the respondent, the method of S. Schwartz “Value Questionnaire” [28], adapted by V. Karandashev [29], was used; for identifying the attitudes to family life and compatibility of family values we used the method of “Role Expectations and Claims in Marriage” by A.Volkova et al. [30]. Additionally to the main methods the study included an express method for assessing marital—the “Marital satisfaction Questionnaire” by V.Stolin et al. [31]. Although the study of the spouses’ satisfaction with their marriage was not the aim of the study, this measurement was added to provide an overall assessment of the effects of consistency or discrepancies in the spouses’ values. The general level of religiosity was also controlled. One item was included in the socio-demographic questionnaire, in which 5 levels of religiosity were briefly described (according to D. Chumakova [24]), and the respondents were asked to choose the description most suitable for them. The sample did not include people who showed radical levels in terms of both involvement in religion and denial of religion. The question of religious affiliation was formulated: “What religion do you belong to?” The respondents indicated their religion themselves. The study included only those respondents who indicated Orthodox Christianity or Islam, according to their ethnic group. Statistical analysis was carried out using descriptive statistics, Mann–Whitney U-test. To assess the degree of statistical significance of the differences between spouses, a paired T-test was used. Calculations were performed using STATIATICA—12. 

Since this is a psychological non-clinical study, it was evaluated in terms of APA “Ethical Principles of Psychologists and Code of Conduct”, date 2003 with latest amendments 2017. The protocol of the study was approved by the Committee on publication ethics (COPE) of Peoples’ Friendship University of Russia (RUDN) (Project identification code—050422-0-010).

## 3. Results

### 3.1. Lfe Values in Inter-cultural Families

A comparative analysis of the life values of spouses from the three considered groups (Table 1) first of all shows that at the normative, cultural level, there are more statistically significant differences than at the level of personal values. In general, this once again confirms S. Schwartz’s theory position about the differences between the normative values of a person and his/her everyday reaction [28,32]. Statistically significant differences at the normative level of values were revealed for all values between Russian and Russian–Arab couples; between Russian–Caucasian and homogeneous Russian couples—for all values except conformance. For spouses from mono-cultural and Russian–Caucasian couples, conformity is less important than for the spouses from Russian–Arab ones. As inter-ethnic–inter-faith couples have a large social distance both between spouses and between one of the spouses and the couple’s social environment, for better social adaptation, they need conformity. In general, most normative values are much more important for Russian–Arab families than for Russian ones, which supports the differences in the value of conformity and indicates a greater commitment of this group to normative life values in general.

Comparison of the normative values of Russian–Caucasian couples with Russian–Arab ones revealed that for Russian–Caucasian couples, many normative values are less significant than for Russian–Arab couples. This supports the idea that a larger cultural distance between the spouses in a couple is associated with a greater commitment of the spouses to normative, general human, universal values. There were no significant differences between these groups in the degree of importance of the values of independence, and achievements, which in both groups are recognized as quite important, and in the value of stimulation, which is of low importance for both groups. The structure of the values does not differ radically. 

S. Schwartz’ value theories postulate, that each of the 10 values correlates significantly with a set of everyday behaviors that is motivated with the value. The sets of behaviors ate largely consistent with the motivational continuum. That is, in most cases, values correlate most positively with their expected behaviors, and most negatively with the behaviors motivated by values motivationally opposed. The gender role expectations may moderate correlations in some domains [32]. Thus, the first part of the method “Normative values” reflects a person’s firm beliefs about the degree of importance of various universal values, the second part “Personal values” reflects a person’s individual priorities or typical behavioral attitudes in certain life situations. This allows us to see the degree of coordination of beliefs and real motives of a person’s behavior, as well as how much a person manages to realize his/her life values in real life (which is especially actual in situations of inter-cultural interaction).

At the level of individual priorities (everyday behavior) between three groups of couples less significant differences were found. There were no significant differences in the importance of individual values of stimulation for all three groups. The value of self-direction is less important for mono-cultural couples than for both types of cross-cultural couples, but especially so in comparison with Russian–Caucasian. 

In comparison with mono-cultural Russian couples, the everyday behavior of spouses from Russian–Caucasian couples is less motivated by the values of benevolence, hedonism and security, but is more strongly motivated by the values of self-direction and power. This is consistent with the ethnic stereotypes of the groups.

The differences between individual values in Russian–Arabian and mono-cultural Russian couples shows the stronger orientation of inter-faith couples for the values of traditions, universalism, power and security. That is also consistent with the ethnic stereotypes.

The differences between individual values in Russian–Caucasian and Russian–Arabian couples are, in general, the same as at the normative level, with one exception. If there was no significant difference in the importance of achievement, then at the normative level for Russian–Arab couples achievement is more important than for Russian–Caucasian, difference in the values of power which is significant at the normative level (with its greater importance for the Russian–Arab couples, at the individual values are leveled.

In the hierarchy of individual values for all three groups, independence (self-direction) is the most important value, followed by security and achievements. On the whole, the life values of couples of Russian women that are representatives of different cultures at the normative level differ more significantly than at the individual level. 

In general, for inter-cultural families, normative (general cultural) values are more important than for mono-cultural ones. Presumably, the greater cultural distance of the spouses in a couple is associated with the greater commitment of the spouses to normative values. Furthermore, for intercultural families in general, the values of self-reliance, stimulation (new impressions) and power are more important than for mono-cultural ones.

The value of achievements, recognized by Russian–Caucasian families as one of the most important at the normative level, does not find its realization in everyday behavior, is considered less important than in other families. Furthermore, for this group, conformity, kindness, hedonism, and security are important in everyday life, and power is more important than others. Apparently, at the individual level, the value of power in some way compensates for the rejection of the value of achievements.

### 3.2. Family Values in Inter-cultural Families

The comparative analysis of the degree of importance of family values for the spouses (Table 2) showed that for the Russian–Caucasian couples, the household sphere is important to a greater extent than for the other two groups, and the parental sphere is more important than for mono-cultural couples. Perhaps this is due to the cultural traditions of the regions where the respondents live. In the Transcaucasian region, a good home improvement, comfort and hospitality are very much appreciated. The Russian–Caucasian group also demonstrated a statistically significant greater willingness of spouses, regardless of gender, to assume obligations in these two areas. It is interesting to note that the strongest differences among Russian–Caucasian families were revealed in claims in the household sphere with Russian families, and in the parental sphere—with Russian–Arab families.

In the expectations from the partners in these two spheres, the Russian–Caucasian families also show significantly higher indicators than Russian—Arab in the two spheres, but significant differences with mono-cultural Russian families on this parameter were not revealed. That is, compared to the Russian–Arab and Russian–Caucasian, homogeneous Russian families consider the householding and parental functions of the family to be less significant, they are less willing to fulfill these roles and to a lesser extent expect it from their partners. For Russian–Arab families, in comparison with Russian–Caucasian, these areas are also less significant, they are less willing to fulfill these roles themselves, but not less than Russian–Caucasian ones expect their performance from their partners.

In addition, Russian–Arab couples showed a higher level of expectation for the partner’s appearance, compared to mono-cultural Russian and Russian–Caucasian couples. And spouses from Russian–Caucasian couples expected less from their partner’s therapeutic role performance, compared to both other groups, especially mono-cultural Russian. Significant differences between the studied groups were also revealed in the role adequacy of the couples (i.e., the degree of conformity of the expectations of one partner from another to the role claims of the second partner, i.e., the extent to which one is willing to take responsibility for performing a particular role). A low degree of role adequacy implies an explicit or latent role conflict. In Russian–Caucasian families, the role adequacy in matters of the partners’ appearance is significantly higher than in Russian (*p* = 0.002), and in the therapeutic sphere—though significantly higher than in Russian–Arab (*p* = 0.03). Comparing these results with the data of our previous study of Russian–Arab families’ values [33], we believe that these data reflect a greater psychological well-being of inter-ethnic couples with a common religion, compared with inter-faith couples; as well as a lesser significance for family life of external, not directly related to the relationship, factors (in particular, the appearance of a partner) in inter-cultural couples, compared with mono-cultural Russian couples.

### 3.3. Consistency of Spouses’ Values

Of particular interest to our study was the degree of consistency of values (value unity) between spouses in families of different types. With the help of paired T-test, we compared the degree of divergence in the ideas of the spouses about the importance of life and family values and attitudes towards family life in the types of families under consideration. When assessing the consistency of values in the couples, the difference between the indicators of husbands and wives was calculated. Accordingly, positive values of the difference variable mean higher levels of importance of a certain value for husbands, and negative values mean higher levels of importance of the value for wives.

The analysis of the degree of coincidence of the life values of spouses showed that statistically significant differences in the value concepts between Russian spouses in homogeneous couples are minimal, and the maximum number of discrepancies was noted in Russian–Caucasian couples. In Russian families, significant differences at the normative level are noted only in the values of universalism (*p* = 0.02) and power (*p* = 0.04). Women are more focused on the universal values, than their husbands, and men - on the value of power, which reflects gender stereotypes. At the individual level, a significant discrepancy exists only in the value of independence (*p* = 0.07), also in accordance with gender stereotypes, for men this value is more important.

At the normative level, Russian–Arab couples significantly differ in their ideas about the importance of conformity values (*p* = 0.009, prevails in women), achievements (*p* = 0.02) and power (*p* = 0.003, prevail in men). At the individual level, differences in the values of power (*p* = 0.00000 *), independence (*p* = 0.0005), safety (*p* = 0.004) and achievements (*p* = 0.02), which prevail in men, are also significant. That is, in this group of families, gender stereotypes are even more pronounced in the value orientations of the spouses, both at the normative and at the individual levels.

In Russian–Caucasian families, significant differences were found in the values of power prevailing in men (*p* = 0.0000 *), achievements (*p* = 0.0001), independence (*p* = 0.01), traditions (*p* = 0.03) and security (*p* = 0.05); in women, compared with their men, the value of kindness prevails (*p* = 0.02). Differences in individual values largely coincide with the normative level, but significant predominance of hedonism value (*p* = 0.05) in women was also found.

Such results suggest that the existence of a common religion in inter-ethnic families increases the importance of traditional values (as was proved above), including in relation to the traditional distribution of social roles between the genders. This, in turn, suggests that the differences in values reflected not so much the contradictions in the values of the spouses as the complementarity of the values of men and women.

Next, let us consider the ratio of family values of spouses from different types of couples. Table 3 provides data on the statistical significance of differences in the views of spouses on the importance of different areas of family life.

As can be seen from Table 3, the minimum number of differences is observed in Russian families, and the maximum in Russian–Arab families. For men in Russian mono-cultural families, as in inter-cultural families, the sexual sphere is more significant than for their wives, which is quite traditional. Besides, for men in Russian families, more than for their wives, the sphere of parenthood is more important. In Russian–Caucasian families, the same difference is noted at the level of significance. It speaks, rather, of the declining significance of the traditionally female parental role in modern Russian women, who are both in mono-cultural and inter-cultural marriages, apparently due to the assumption of new roles, as well as to the distribution of parental responsibilities with their husbands. However, in marriages with Muslims—in comparison with marriages with Christians—Russian women retain a greater importance of the parent sphere compared with their husbands.

Interestingly, in Russian–Caucasian families, the household sector and the appearance of their partners is more important for men than for their wives; whereas in Russian–Arab families, on the contrary, these areas are much more important for women. There are also significant differences in the views of spouses from Russian–Arab families on the importance of the psychotherapeutic sphere of the family, which is more significant for women. Social activity is also more important for women in these families. Perhaps these differences are explained by the fact that the families participating in this study lived in the motherland of women, the men were in a foreign cultural environment, which could reduce the importance of social activity for them.

In general, these results show that a large cultural distance between spouses is associated with large differences in the ideas of spouses about the importance of areas of family life. Moreover, the factor of common religion reduces the number of contradictions.

As can be seen from Table 4, in mono-cultural Russian families there are no significant differences in the role claims of the spouses. Both men and women are equally willing to take responsibility for all major areas of family life. There are significant differences in both types of inter-cultural families. As described above, features in the distribution of roles between spouses were discovered. Thus, in Russian–Caucasian families, the overall level of role-playing claims is higher for men than for their wives, and in Russian–Arab ones it is higher for women. The same differences are noted in relation to the psychotherapeutic role. Furthermore, in Russian–Caucasian families, men are much more willing to implement an external social activity, while in Russian–Arab families there are no significant differences in this role—in Russian–Arab families, women are more willing to fulfill household roles and to take care of their appearance. Partially, these data are explained by gender stereotypes, but it is also obvious that a lot determines the place of residence, i.e., which of the spouses lives in a foreign environment [1].

As can be seen from Table 5, in mono-cultural Russian families there are also no significant differences in the expectations of the spouses in relation to each other. In Russian–Arab families, the only difference was revealed—that women expect more from their husbands to realize external social activity than their husbands from them. As already mentioned, we link this is due to the fact that men in these families are in a foreign cultural environment, which apparently reduces their social activity (while maintaining the desire to assume this role), which causes, perhaps, some tension in the expectations of their wives in this domain. In Russian–Caucasian families there are differences in most areas of family life. Men expect more from their wives to perform household roles and social activity, women have a higher overall level of role expectations in relation to their spouses. That also confirms the previously described influence of cultural background on women in these families. Expectations for the spouses’ appearance in these families are higher for women. As mentioned earlier, this area is more meaningful for women from these families; therefore, apparently, they expect their husbands to meet their demands regarding appearance.

Thus, we see that in inter-cultural families there are more differences in the perceptions of spouses about the importance of the spheres of family life and the distribution of roles between spouses. On the one hand, these discrepancies reflect traditional stereotypes about genders’ social and family roles. On the other hand, the factor of foreign cultural environment plays an important role in the attitudes toward family life of spouses from inter-cultural families. The shared religion of spouses contributes to greater consistency of family values of spouses but also contributes to a greater commitment of spouses to traditional family life patterns.

### 3.4. Marital Satisfaction

Next, we will consider how the identified features in the values and the degree of their consistency manifest in terms of satisfaction with marriage (see Table 6). We evaluated both the satisfaction with the marriage of each spouse, and the average satisfaction of the couple, as well as the degree of consistency of the spouses about the welfare of their marriage. As can be seen from Table 6, a pair-wise comparison of the three types of couples showed differences in all parameters of marital satisfaction at a very high level of statistical significance. In inter-ethnic families with a common religion, marital satisfaction was significantly higher than in other families. In inter-ethnic and—at the same time—inter-faith couples, the level of marital satisfaction was the lowest. Furthermore, in the Russian–Caucasian families the lowest coefficient of differences was in the ideas of the spouses about the welfare of their marriage—meaning they had the highest level of agreement. By contrast, in homogeneous Russian families, this indicator is the highest, which suggests a low degree of openness in marital communication, and the fact that marital satisfaction in this group is determined not so much by the quality of relationships, as by lower expectations in relation to family life.

## 4. Discussion

Summing up the general results of the study, we can draw the following conclusions. In general, an inter-ethnic family is a complex structure in which different values and relationships not only compete, coexist but also transform, change the national and cultural identity and cultural values of the spouses. Different religions of spouses create additional difficulties for establishing interaction in a couple, these difficulties are associated not so much with life values, as with family attitudes, the distribution of roles in the family.

In general, for inter-cultural families, normative (general cultural) values are more important than for mono-cultural ones (including spouses belonging to a common ethnos and religion). The greater the cultural distance of the spouses in a couple, the more the spouses are committed to normative values.

The family values of inter-ethnic couples with a common religion, compared to inter-faith couples, are more traditional in terms of the gender distribution of roles. Compared to mono-cultural families, both types of inter-cultural ones are more focused on external factors (for example, the appearance of partners). The degree of role adequacy (i.e., the conformity of role expectations to a partner and his willingness to perform certain roles) do not significantly differ in inter-cultural and mono-cultural families.

In inter-cultural families, the degree of coincidence of the life values of spouses is much less pronounced than in mono-cultural ones. But in couples practicing a common religion, differences in life values reflect not so much the contradictions in the values of spouses, as the complementarity of the values of men and women.

In family values and role attitudes of spouses from inter-cultural marriages, there are also significantly more differences than in mono-cultural couples. On the one hand, these discrepancies are also associated with adherence to traditional values, on the other hand, in the family attitudes of spouses from inter-cultural families, the factor of one of the spouses’ living in foreign culture environment plays an important role.

## 5. Conclusions

The study was aimed at analyzing the specifics of the hierarchy and the degree of life and family values consistency in inter-cultural couples of two different types: inter-ethnic and inter-faith. We also pursued the aim of assessing differences in the degree of general marital satisfaction of the spouses in different types of inter-cultural couples. The need for such a study is dictated by the widespread ideas that religious differences of spouses are the main obstacle to inter-cultural marriage, as well as the main risk factor for the relationships quality in such a marriage, and, at the same time, by insufficient empirical evidence of these ideas. A comparative analysis of two groups of inter-cultural couples was conducted. As an example of inter-faith couples, couples consisting of Russian Orthodox Christian women and Arab Muslim men were taken. As an example of inter-ethnic couples with a common religion, couples consisting of Russian women with the same characteristics—as in the first group—and men from the Transcaucasian region belonging to the Christian religion were chosen. Each group was also compared with a control of mono-cultural couples, consisting of ethnically Russian Christian men and women. In cross-cultural studies, it is almost impossible to select a sample in one culture so that it exactly matches the group in another culture. Matching on one variable, with almost no exceptions, leads to a mismatch on other variables [34] p. 326. Therefore, cross-cultural research should carefully generalize its results, transferring them to some larger populations [35]. Thus, when disseminating the results of this study, it is important to take into account that, firstly, the data were obtained in groups with a certain combination of certain cultures, and secondly, the purpose of the study was not to study these specific cultures, but to study some common patterns of inter-cultural marriage on their example. That allows us to better understand the psychological mechanisms for the implementation of cultural and personal values in family relationships; about the impacts of single cross-ethnic factor or combined cross-ethnic and cross-faith factors to the processes of mutual adaptation (and acculturation) in inter-cultural families, about the differences in the value adjustment between spouses from these types of couples.

According to the results obtained, in general, for inter-cultural couples, normative values are more important than for mono-cultural ones. The greater the cultural distance of the spouses in a couple, the more the spouses are committed to normative values. The study, contrary to popular belief, showed no radical differences in the life or family values of spouses in inter-cultural couples. The identified differences did, however, reflect gender stereotypes about the distribution of family roles.

The presence of a common religion in an inter-ethnic couple contributes to greater marital satisfaction and greater consistency in the representations of the spouses about the welfare of their relationship. In inter-ethnic–inter-faith couples, marital satisfaction is the lowest, but the degree of the consistency of spouses about the quality of relationships is higher than in mono-cultural families, indicating a more open communication in these families and the involvement of spouses in relationships. That is, the affiliation of spouses to different cultures (both ethnic and religious) optimizes the process of comparing values and family attitudes. Despite a number of difficulties, spouses from inter-cultural families, in general, have more consistent ideas about their family life, that is usually connected with more secure attachment in a couple.

The research data can be used by psychologists and social services to help cross-cultural families in realizing their potential more fully and overcoming the limitations identified in the study. In particular, more open communication and greater consistency in their family life representations of both types of cross-cultural families can be viewed as a resource to support a foreign spouse in enhancing his or her contacts with the social environment and in better implementation of his or her life values into everyday behavior. The fact that the spouses with a greater cultural distance combine their discrepancy in life and family values with high marriage satisfaction makes it possible to use cultural differences in a couple as a resource for constructively distributing family roles and optimizing family and personal functioning as a whole.

The study has a number of limitations associated with the prospects for further research on this problem. 

Firstly, the sample of the study was not so big; however, this is partially offset by the fact that the three subgroups have been carefully selected and equalized to each other according to the main socio-demographic characteristics due to their selection from a large number of respondents.

Due to the limited volume, the discussion of the results is limited. A more detailed comparison of the data of this study with other studies of inter-cultural couples is required. Furthermore, since—as mentioned above—the parameter of the religion of spouses is rarely taken into account in modern cross-cultural studies of couple relationships; therefore, additional, similar studies are required involving samples with other combinations of ethnic and religious characteristics.

In this study, we compared three groups in which women had similar characteristics, and men’s ethnic and religious characteristics varied. To better understand the role of the gender factor, it is advisable to conduct a similar study, which will compare the marriages of men with the same characteristics and women with varying ones.

The disadvantage of this study is that the groups were not aligned according to their place of residence (native or foreign environment). In one of the inter-cultural subsamples (Russians–Arabs), the men were emigrants, in another (Russians–Caucasians), the women were emigrants. We were forced to do this to strengthen the distinguishing ethnic and religious characteristics. But some results obtained can be explained precisely by the factor of living in a foreign environment. Accordingly, more research is needed to assess the role of this factor.

Factors of spouses’ ethnic identity, the level of their religiosity and religious involvement, the degree of social adaptation in society also deserve special attention. Our study showed that all these factors can moderate the process of harmonization of value structure in a couple, and the interaction of the spouses’ value unity with their marital satisfaction. It would also be useful to investigate the attitudes and relations of cross-cultural spouses at different stages of the family’s life cycle.

## Figures and Tables

**Table 1 behavsci-10-00038-t001:** Comparison of the importance of life values for spouses of mono-cultural, inter-ethnic and inter-faith couples.

Life Values	Types of Couples						Statistical Significance of Differences		
	Russian– Caucasian		Russian– Arab		Russian– Russian		R–C/ R–R	R–A/ R–R	R–C/ R–A
	Mean	SD	Mean	SD	Mean	SD	*p*	*p*	*p*
*Normative Values*									
Conformity	3.23	1.09	4.53	0.64	2.86	0.87	0.094	**0.000**	**0.000**
Tradition	3.69	0.69	4.62	0.79	2.44	0.91	**0.000**	**0.000**	**0.000**
Benevolence	4.30	0.67	4.91	0.75	3.76	0.72	**0.001**	**0.000**	**0.000**
Universalism	3.96	1.17	4.43	0.70	3.03	0.63	**0.000**	**0.000**	**0.013**
Self-Direction	4.91	0.75	4.91	0.72	3.87	0.71	**0.000**	**0.000**	0.986
Stimulation	3.17	1.24	3.15	0.93	2.44	0.94	**0.004**	**0.000**	0.943
Hedonism	3.63	0.86	4.38	1.20	3.07	0.95	**0.009**	**0.000**	**0.001**
Achievement	5.03	0.83	4.77	0.76	3.29	0.75	**0.000**	**0.000**	0.146
Power	3.73	0.93	4.28	1.01	2.58	1.15	**0.000**	**0.000**	**0.015**
Security	4.41	0.91	5.49	0.49	3.46	0.79	**0.000**	**0.000**	**0.000**
*Personal Values*									
Conformity	1.10	0.69	2.27	0.74	1.74	0.76	**0.0002**	**0.001**	**0.000**
Tradition	1.11	0.61	2.10	0.79	0.90	0.66	0.156	**0.000**	**0.000**
Benevolence	1.52	0.69	1.94	0.82	1.83	0.62	**0.042**	0.441	**0.009**
Universalism	1.38	0.69	2.17	0.59	1.36	0.48	0.872	**0.000**	**0.000**
Self-Direction	2.41	0.59	2.31	0.60	2.11	0.59	**0.037**	0.131	0.461
Stimulation	1.30	0.68	1.37	0.66	1.56	1.04	0.198	0.259	0.629
Hedonism	0.96	0.49	1.88	1.08	1.88	1.04	**0.0000**	0.980	**0.000**
Achievement	1.72	0.65	2.14	0.81	1.91	1.04	0.356	0.233	**0.013**
Power	1.61	0.56	1.58	0.99	1.12	0.87	**0.008**	**0.040**	0.890
Security	1.71	0.45	2.82	0.62	2.18	0.88	**0.006**	**0.000**	**0.000**

**Table 2 behavsci-10-00038-t002:** Comparison of the importance of family values for spouses of mono-cultural, inter-ethnic and inter-faith couples.

Life Values	Types of Couples						Statistical Significance of Differences		
	Russian– Caucasian		Russian– Arab		Russian– Russian		R–C/ R–R	R–A/ R–R	R–C/ R–A
	Mean	SD	Mean	SD	Mean	SD	*p*	*p*	*p*
*The importance of the family spheres*									
Sexual	5.84	1.26	5.55	2.12	5.60	1.52	0.462	0.904	0.475
Personal identification with the spouse	6.53	1.15	6.47	1.89	6.65	1.54	0.691	0.614	0.864
Householding	6.46	1.12	5.77	1.33	5.50	1.43	**0.002**	0.350	**0.012**
Parental	6.91	0.83	6.49	1.57	6.38	1.16	**0.028**	0.696	0.144
Social activity	6.66	0.69	6.48	1.48	6.41	1.62	0.385	0.825	0.513
Psychotherapeutic	6.61	0.95	6.99	1.46	6.86	1.37	0.338	0.663	0.161
Appearance	6.16	0.96	6.37	1.95	6.26	1.44	0.710	0.783	0.565
*Role claims in the family spheres*									
Householding	6.42	1.36	5.60	2.00	5.60	1.88	**0.048**	0.128	**0.033**
Parental	6.95	1.27	6.08	2.24	6.28	2.10	**0.034**	0.225	0.091
Social activity	6.92	0.67	7.03	1.71	6.78	1.91	0.715	0.797	0.669
Psychotherapeutic	6.32	1.14	6.20	2.31	6.53	1.39	0.785	0.059	0.472
Appearance	5.95	1.23	6.28	2.24	5.48	1.99	0.452	0.118	0.211
*Role expectations from the partners in the family spheres*									
Householding	6.50	1.38	5.93	1.71	5.40	1.88	0.106	0.272	**0.002**
Parental	7.05	0.92	6.90	1.78	6.48	2.10	0.627	0.487	**0.050**
Social activity	6.39	0.99	5.93	1.98	6.05	1.91	0.268	0.359	0.335
Psychotherapeutic	6.87	1.09	7.78	1.34	7.20	1.39	**0.001**	0.092	0.314
Appearance	6.39	0.93	6.45	2.30	7.10	1.99	0.890	**0.011**	**0.023**

**Table 3 behavsci-10-00038-t003:** The consistency of the spouses’ attitudes about the significance of the family spheres in the couples of different types.

Family Life Spheres	Types of Couples								
	Russian–Caucasian			Russian–Arab			Russian–Russian		
	Diff. *	t	*p*	Diff. *	t	p	Diff. *	t	*p*
Sexual	0.74	3.68	**0.0017**	1.70	3.58	**0.0012**	0.80	2.56	**0.0193**
Personal identification with the spouse	0.32	1.03	0.3163	−0.60	−1.61	0.1188	−0.30	−1.19	0.2492
Householding	0.82	3.55	**0.0023**	−0.80	−2.23	**0.0339**	0.00	−0.01	0.9901
Parental	0.34	1.95	0.0669	−0.52	−1.25	0.2215	0.65	2.20	**0.0401**
Social activity	0.11	0.70	0.4944	−0.90	−2.79	**0.0093**	−0.13	−0.44	0.6663
Psychotherapeutic	0.26	1.17	0.2557	−0.75	−2.27	**0.0311**	−0.13	−0.42	0.6804
Appearance	0.58	3.54	**0.0023**	−1.23	−2.52	**0.0174**	−0.23	−0.55	0.5891

* positive values of the difference variable mean higher levels of importance of a certain value for husbands, and negative values mean higher levels of importance of the value for wives.

**Table 4 behavsci-10-00038-t004:** The consistency of role claims (willingness to assume responsibility for the areas of family life) of spouses in different types of couples.

Family Life Spheres	Types of Couples								
	Russian–Caucasian			Russian–Arab			Russian–Russian		
	Diff. *	t	*p*	Diff. *	t	p	Diff. *	t	*p*
Householding	0.42	0.88	0.3915	−2.47	−4.20	**0.0002**	−0.50	−0.85	0.4085
Parental	0.32	0.78	0.4457	−0.57	−0.90	0.3759	1.20	1.95	0.0655
Social activity	1.21	5.11	**0.0001**	0.73	1.65	0.1097	0.45	1.34	0.1965
Psychotherapeutic	0.53	2.14	0.0465	−1.60	−3.72	**0.0009**	−0.15	−0.38	0.7112
Appearance	0.11	0.62	0.5416	−2.17	−3.57	**0.0013**	−0.25	−0.40	0.6949
General level of claims	0.52	3.15	**0.0055**	−1.21	−3.17	**0.0036**	0.22	0.82	0.4204

* positive values of the difference variable mean higher levels of importance of a certain value for husbands, and negative values mean higher levels of importance of the value for wives.

**Table 5 behavsci-10-00038-t005:** The consistency of role expectations (expectations from the spouse of taking responsibility for the spheres of family life) of spouses in different types of couples.

Family Life Spheres	Types of Couples								
	Russian–Caucasian			Russian–Arab			Russian–Russian		
	Diff. *	t	*p*	Diff. *	t	*p*	Diff. *	t	*p*
Householding	1.11	3.33	**0.0038**	0.87	1.85	0.0749	0.50	1.03	0.3145
Parental	0.42	1.80	0.0880	−0.47	−1.08	0.2905	−0.25	−0.53	0.6005
Social activity	−1.00	−5.85	**0.0000**	−2.53	−5.07	**0.0000**	−0.70	−1.35	0.1929
Psychotherapeutic	0.26	0.86	0.3986	0.10	0.29	0.7773	−0.10	−0.30	0.7663
Appearance	1.21	4.65	**0.0002**	−0.30	−0.52	0.6075	−0.10	−0.26	0.8004
General level of expectations	0.40	3.17	**0.0053**	−0.47	−1.61	0.1186	−0.13	−0.56	0.5793

* positive values of the difference variable mean higher levels of importance of a certain value for husbands, and negative values mean higher levels of importance of the value for wives.

**Table 6 behavsci-10-00038-t006:** Marital satisfaction in mono-cultural, inter-ethnic and inter-faith couples.

Marital Satisfaction Rates	Types of Couples M (SD)			Statistical Significance of Differences (*p*)		
	Russian–Caucasian (RC) (n = 40)	Russian–Arab (RA) (n = 60)	Russian–Russian (RR) (n = 38)	RC/RA	RR/RA	RC/RR
Individual marital satisfaction of the spouses	39.92 (2.8)	24.23 (5.3)	29.52 (13.0)	**0.00010**	**0.0056**	**0.00000**
Average marital satisfaction in couples	39.95 (2.5)	24.23 (3.0)	29.32 (9.6)	**0.0000**	**0.0002**	**0.00000**
Discrepancies in the spouses’ marital satisfaction inside couple	1.74 (1.7)	7.07 (4.8)	12.95 (11.1)	**0.0000**	**0.0004**	**0.00000**

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
