# Peer review of "Life and Family Values Similarity in Inter-Ethnic and Inter-Faith Couples"

_behavsci, 2020, doi:10.3390/bs10010038_

Round 1

Reviewer 1 Report

See Review Comments file attached.

Author Response

Thank you very much for you comments. We tried to correct the manuscript in accordance with them. 

Please see the attachment for the responses foe each point.

Reviewer 2 Report

I very much appreciate the effort and attention of this article to examine questions of inter-religious and inter-ethnic couples.  the paper does a nice job of shaping the inquiry, and trying to account for all of the various aspects of marital life and satisfaction that might be at play in the formation of a couple's values (both shared and not).

The research design, therefore, is solid (though the sample is small).  there are, however, a number of ways in which it could be improved both for impact and for accessability.

1.  The langauge is cumbersome, at times, and obscure at others.  There are moments in which the authors' efforts to explain everything all at once actually get in the way of their desire to explain the phenomena they are observing.  For example, line 177-178 are pretty hard to understand on their own, without some checking of the table that follows, and some re-reading of preceding material.

2. some of this might be mitigated by reproducing the scales in an appendix or something, so that the discussion of all of hte measures can be clearer to your readers. 

3.  You reach a conclusion about gender, but the gender dimension of your findings are mostly presented in the narrative, rather than in tables themselves.  If gender is a significant factor, it would probably benefit the paper to present a table that lays out the gender differences for your readers, particualrly in a circumstance when you have a nice male-female representation (because you're looking at married couples). 

4.  Your conclusions are a little to general - they're not universailly true, of course, but you generalize a little too liberally.  Even just repeating "in our sample," or "among our study subjects" would remind readers that you're not talking about all inter-ethnic couples ,but that you are talking about specific kinds of couples in a speciifc socio-historical context.

5. On that same score, you might want to offer a little more insight into questions of the similarities or differences of "ethnic" and "religious."  what makes a difference "ethnic" and what makes one "religious?"  Specifically, in the context of Russian couples, where religion and ethnicity are often applied in inconsistent ways (I'm thinking, for example, of Jews, though the same may be true of "arabic," which sometimes might mean "muslim," but could just as easily mean "christian."). You seem to treat the two concepts as clear and distinct, and they are not.  You may, ultimately treat them as such, but you should be careful about how you use those terms, as they matter a great deal and mis-applying them -- or attributing a certain finding to "ethnicity" when it can be better explained by religion -- may lead to inaccurate conclusions.

6.  that said, I think you have some interesting findings here, that could be useful to other researchers in the field. and I encourage you to make these changes: to sharpen your terms and temper your conclusions.

Author Response

Thank you for your attention to our study and for the positive assessment of it and valuable recommendations!

1) & 2) We have already corrected some hard -to-read parts of the text in accordance with the recommendations of the first reviewer. When we make all the changes recommended by the reviewers in the text, we will use the services of the journal to check the language.
Initially, the lack of some data tables and insufficiently detailed comments were due to the limitation of the article to 8 pages. As it became possible to increase the volume, tables and more detailed comments were added in the revised version of the text.

3) To reduce the volume of tables 3-5, we omitted the mean values for men and women, but indicated the differences between them.

When assessing the consistency of values in the couples, the difference between the indicators of husbands and wives was calculated. Accordingly, positive values of the difference variable mean higher levels of importance of a certain value for husbands, and negative values mean  higher levels of importance of the value for wives..

Now we  added an explanation about this in the text and notes to tables 3-5

4) In the revised version we commented that points and wrote about it in limitations and perspectives section

5) added

6) Done.

Reviewer 3 Report

The topic of the paper is interesting, but I think that the econometric analysis can be improved. 

My first concern is about the estimation of the differences. It is not clear from the text whether the demographic and economic characteristics of the participants to the survey are controlled for. The analysis would remain descriptive, but it would assure that the origin of the differences is actually cultural and do not depend on observable characteristics. 

Second, Schwartz & Mare (2005) and Furtado & Theodoropoulos (2011) provide some examples of alternative methods to measure assortative matching. Would it be possible to apply them to your analysis? If not, why? If yes, are the results still robust? 

Third, are the results comparable to similar analysis run on Census data (hence larger samples)? 

Fourth, is there any way to exclude or limit reverse causality concerns? The decision to marry is endogenous, but, for example, the place of birth is less endogenous (it is taken by the parents). Would the differences remain significant if the reference is not the culture or religion of the spouses but some of their parents' characteristics?

Author Response

Thank you for your kind attention and very useful questions and recommendations!

1) 

These subsamples were formed from a much larger number of respondents 225 couples (450 persons), including 85 Russian - Arab; 80 Russian - Russian and 60 Russian - Caucasian. This helped us to equalize subsamples that were consistent with each other by most basic socio-demographic characteristics: age, education level, type of work, children, previous divorces, children from previous marriages, self-rating level of material well-being and so on.

The age of the respondents was from 23 to 47 years (M = 36: women – 32; men – 40), they have been married from 3 to 24 years (M = 6.3); for 30% of respondents this marriage was the second; the majority (87%) of the surveyed families had 1-2 children, aged from 1 to 13 years old (M = 5.2); 10% of couples had children from previous marriages, everyone has a job (full-time), mainly hired employees, ordinary employees or mid-level managers, foreign partners lived in their spouse's country from 5 to 30 years (M = 9.2), everyone were moderately religious.

In the groups of intercultural marriages, unlike monocultural, there were no couples with unregistered marriages (in the monocultural group, 30% cohabitation); in the group of Russian-Caucasian marriages, unlike the other 2 groups, women on average occupy lower-level positions at work. We believe that the absence of cohabitation is related to cultural traditions and migration issues. A lower level of the job position of women married to a Caucasian partner is due to the factor of their living in a more traditional culture.

We added more detailed description of the sample to the paper.

2)  Thank you for the idea to use the contingency table analysis, we think it would be useful. Now we don’t have enough time to try it for including the results into this manuscript, but will use it in next studies.

3) The main focus of our study was the values of the respondents and the degree of consistency in the couple. Unfortunately, the Census in Russia and in Transcaucasian countries does not include this aspect. Our data regarding the value orientations of Russians and Arabs are consistent with S. Schwartz cross-cultural values studies, but Arab countries are poorly represented in that, and the Transcaucasian region is not represented at all. We have not been able to find comparative studies of just such a combination of ethno-cultural samples.

4)  Reverse and multiple causality is common problem for cross-cultural studies. In this study we tried to limit it by more careful alignment of the samples according to socio-demographic characteristics, assuming that these characteristics, in turn, are due to the influence of the parents.

We are currently working on a project dedicated specifically to the analysis of transgenerational factors role in intercultural marital relations. To do this, we conduct an interview about the history of the extended family.

Round 2

Reviewer 1 Report

Much improved - thank you for the changes, which I feel make the paper much more engaging. 

Reviewer 3 Report

I am fine with the responses to my comments.